# Golgi Apparatus Target Proteins in Gastroenterological Cancers: A Comprehensive Review of GOLPH3 and GOLGA Proteins

**DOI:** 10.3390/cells12141823

**Published:** 2023-07-11

**Authors:** Sandica Bucurica, Laura Gaman, Mariana Jinga, Andrei Adrian Popa, Florentina Ionita-Radu

**Affiliations:** 1Department of Gastroenterology, “Carol Davila” University of Medicine and Pharmacy Bucharest, 020021 Bucharest, Romania; sandica.bucurica@umfcd.ro (S.B.); florentina.ionita-radu@umfcd.ro (F.I.-R.); 2Department of Gastroenterology, “Carol Davila” University Central Emergency Military Hospital, 010825 Bucharest, Romania; 3Department of Biochemistry, “Carol Davila” University of Medicine and Pharmacy Bucharest, 020021 Bucharest, Romania; laura.gaman@umfcd.ro; 4Student of General Medicine, “Carol Davila” University of Medicine and Pharmacy Bucharest, 020021 Bucharest, Romania; andreiadrianpopa@stud.umfcd.ro

**Keywords:** Golgi apparatus, GOLPH3, GOLGA proteins, protein sorting, Golgi morphology, tumor progression, therapeutic targets

## Abstract

The Golgi apparatus plays a central role in protein sorting, modification and trafficking within cells; its dysregulation has been implicated in various cancers including those affecting the GI tract. This review highlights two Golgi target proteins, namely GOLPH3 and GOLGA proteins, from this apparatus as they relate to gastroenterological cancers. GOLPH3—a highly conserved protein of the trans-Golgi network—has become a key player in cancer biology. Abnormal expression of GOLPH3 has been detected in various gastrointestinal cancers including gastric, colorectal and pancreatic cancers. GOLPH3 promotes tumor cell proliferation, survival, migration and invasion via various mechanisms including activating the PI3K/Akt/mTOR signaling pathway as well as altering Golgi morphology and vesicular trafficking. GOLGA family proteins such as GOLGA1 (golgin-97) and GOLGA7 (golgin-84) have also been implicated in gastroenterological cancers. GOLGA1 plays an essential role in protein trafficking within the Golgi apparatus and has been associated with poor patient survival rates and increased invasiveness; GOLGA7 maintains Golgi structure while having been shown to affect protein glycosylation processes. GOLPH3 and GOLGA proteins play a pivotal role in gastroenterological cancer, helping researchers unlock molecular mechanisms and identify therapeutic targets. Their dysregulation affects various cellular processes including signal transduction, vesicular trafficking and protein glycosylation, all contributing to tumor aggressiveness and progression.

## 1. Introduction

The Golgi apparatus also known as the Golgi complex or Golgi body is an organelle that is found within most eukaryotic cells and was discovered in 1898 by the scientist Camillo Golgi using his own metal staining technique called the black reaction, “La reazione nera”. At the time Golgi discovered the Golgi apparatus, he was involved in a great debate regarding neural theory with another scientist, Santiago Ramón y Cajal. Golgi created “The reticular theory” after observing a vast network of neurons in grey matter. In accordance with this theory, the entire cerebrospinal axis functions as a single continuous neural network. The core idea of how the entire nervous system functions is reflected by this theory, yet the reality is very different from what Camillo Golgi suggested. Using the Golgi staining method, Santiago Ramón y Cajal studied the neural network in 1887 and made a finding that would fundamentally alter how we think about the nervous system. He found that there is a space between the neurons rather than a continuous connection between them; this space is now referred to as the synaptic cleft. This is when “The neuron theory” first emerged [1].

The Golgi complex has several roles such as processing, sorting and post-translational modification of proteins and lipids being synthesized in the endoplasmic reticulum as well as other cellular processes [2]. Regarding the structure, the Golgi apparatus is formed by stacked cisternae containing several enzymes that continuously recycle, leading to the maturation of the cisternae involved in transport of cargo through the Golgi complex. The stacked cisternae have a polarized structure with a cis-face and a trans-face connected by membrane vesicles and tubules. Even though the models of transport remain unclear, the cisternal maturation is certainly one of the most important mechanisms that lead to moving the cargo in the Golgi apparatus, which was closely studied, using electron microscopy, by important scientists like Marilyn Farquhar and George Emil Palade [2,3].

## 2. Golgi Protein Roles and Implications

The Golgi body is involved in post-translational modification of proteins previously synthesized in the endoplasmic reticulum, including glycosylation, phosphorylation and proteolysis, glycosylation being the most relevant in our case, but for any of these phenomena to happen there is a population of proteins called resident proteins of the Golgi apparatus which are involved in vesicle formation and transport and protein sorting and are also enzymes [2,4]. Glycosylation is a complex sequential enzymatic effect that forms glycosidic linkages using numerous glycotransferases in order to bind saccharides to other saccharides, proteins or lipids. There are ten monosaccharides that are involved in the formation of glycans: galactose, glucose, N-acetyl-galactosamine, fucose, N-acetylglucosamine, mannose, xylose, sialic acid and glucuronic acid. Glycoproteins are formed by linking one or more glycans that are bound using oxygen or nitrogen linkages, resulting in O-glycans and N-glycans. Mucin-type O-glycosylation, one form of protein O-glycosylation, involves initial attachment of Gal-NAc as the initial monosaccharide that links to serine or threonine residues. In addition to the mucin-type O-glycosylation described above, other structures for O-glycans can take different forms; O-mannose O-linked structures as well as nucleocytoplasmic O-linked b-N-acetylglucosamine structures come to mind as examples for O-linked O-glycans [5,6,7]. Changes can occur in the process of glycosylation and, as a result, an alteration of the process can lead to oncogenic transformations.

Sugar addition at the Golgi level depends on specific nucleotide sugar transporters and glycosyltransferases (GT). Genetic variations in these components have been identified in humans with congenital disorders of glycosylation (CDGs), an umbrella grouping first described by Jaeken and Matthijs in 2001 [8]. However, an increasing subset of CDG disorders are having indirect ramifications on glycosylation pathways by altering Golgi structure, trafficking mechanisms, and homeostasis mechanisms in ways that impact glycosylation pathways. As of 2014, over 110 different CDGs had been identified; approximately one-third impacted Golgi glycosylation as noted by Jaeken in 2015. Screening for CDGs usually involves conducting the transferrin isoelectric focalization pattern diagnostic test which only detects patients with N-glycosylation defects [9].

Tumor cells often exhibit numerous glycosylation alterations, particularly protein glycosylation alterations that contribute to their altered functionality as well as heterogeneity within tumor cells. Heterogeneity often results from protein and cell glycosylation aberrations [7]. The addition of glycans to proteins is an integral post-translational process in tumor onset and development; specifically, it impacts cell proliferation and development and it is known that one-fifth of structural proteins contain glycoproteins [10].

Congenital disorders of glycosylation (CDGs) refer to a collection of disorders caused by mutations to genes responsible for glycosylation pathways in the Golgi apparatus, specifically glycosyltransferases and enzymes involved with Golgi function that control glycosylation of proteins in the Golgi apparatus. Mutations affecting these genes may impact proteins associated with nucleotide sugar transport, glycosyltransferases, enzymes or factors essential to Golgi function which cause abnormal protein glycosylation leading to developmental delay, intellectual disability seizures or abnormal bleeding symptoms [11].

Studies show that mutations in genes responsible for glycosylation pathways in the Golgi apparatus can play a part in cancer formation. Glycosyltransferases and other Golgi-related proteins encoded by these mutations have been seen in different cancer types including gastric, colon and breast cancers, their mutations altering protein glycosylation patterns which then facilitate tumor growth and metastasis [11].

A relevant key player in glycosylation, according to Senitiroh Hakomori, is represented by the glycolipid GM3; GM3 binds with antimetastatic membrane proteins CD9 and CD82 (also known as “tetraspanin”) and impedes cells motility and invasiveness in colorectal cancer. Specifically, this occurs with α-three integrin -CD9–GM3 complexes formed within glycolipid-enriched microdomains which leads to the suppression of Matrigel motility stimulation [12].

Malignancies often exhibit changes to glycosylation that increase branching of 6GlcNAc side chains on N-linked structures, which is driven by upregulated activity of an enzyme called GnT-V promoted by the Ets oncogene family. GnT-III synthesizes 4GlcNAc (bisecting GlcNAc), thus counteracting any branching. The balance between GnT-V and GnT-III ultimately determines epitope levels on cells.

Upregulation of the GnT-V gene leads to enhanced 6GlcNAc branching, which has a pro-metastatic effect. On the other hand, the GnT-III gene reduces 6GlcNAc branching and exhibits antimetastatic properties; its inhibitory effect on 6GlcNAc branching inhibits metastasis in B16 melanoma. E-cadherin undergoes reduced 6GlcNAc branching while increasing 4GlcNAc formation, resulting in increased cadherin-dependent cell adhesion and suppression of metastasis [12].

Notably, GnT-V’s pro-metastatic effect can be attributed to its stabilization of active matriptase through the addition of a 6GlcNAc side chain.

Mutations of genes involved in glycosylation pathways in the Golgi apparatus can both cause developmental disorders and contribute to specific forms of cancers [7,9]. Defects in glycosylation pathways can have serious repercussions for both disease development and progression, potentially leading to abnormal protein glycosylation with congenital disorders of glycosylation and possibly contribute to some types of cancer such as gastrointestinal cancers which are the main focus of this review [7,9].

## 3. Genetic Abnormalities in Genes Directly Involved in Glycosylation Pathways

Multiple genes involved with Golgi glycosylation pathways have been implicated in gastric cancer. The most common ones are as follows:

ST6GALNAC1: This gene encodes an enzyme which adds sialic acid to O-linked glycans. Its expression tends to be higher in gastric cancer tissues compared with normal tissues and has been linked with tumor progression and metastasis [13].

MGAT5: This gene encodes for an enzyme responsible for adding N-acetylglucosamine (GlcNAc) residues to N-linked glycans, with increased overexpression being linked with increased invasiveness and poor prognosis in gastric cancer tissues. Studies have also linked this overexpression with increased invasiveness and poor prognosis [14,15].

B4GALT1: This gene encodes an enzyme responsible for adding galactose to N-linked glycans, with increased expression levels being seen in gastric cancer tissues, suggesting its involvement with tumor progression and metastasis [16].

Other genes involved in glycosylation pathways could also play a part in gastric cancer, but more research must be conducted in order to ascertain their precise roles in this type of disease.

## 4. Genetic Abnormalities in Genes Indirectly Involved in Glycosylation Pathways: Vesicular Transport

Disruptions to genes related to vesicular transport may alter Golgi function and cause abnormal protein glycosylation. Proper functioning of the Golgi apparatus relies on accurate movement of vesicles between its cisternae and organelles via proteins and pathways; mutations within this network can alter Golgi structure and function and ultimately alter protein glycosylation patterns [17].

Examples of genes involved in vesicular transport that have been linked with CDG disorders and abnormal glycosylation include COG complex proteins, GRASP55 and syntaxin-5. Mutations affecting these genes may result in various CDG syndromes such as CDG-IIj, CDG-IIh or CDG-IIa disorders [17,18,19].

Overall, mutations in genes indirectly associated with glycosylation—specifically those involved with vesicular transport—can have serious repercussions for Golgi function and protein glycosylation.

Aberrant glycosylation in the Golgi apparatus has been shown to regulate cancer cell invasion across a range of cancer types, including prostate, breast and gastric cancers. Golgi glycosylation dysregulation has been implicated in various molecular and cellular processes involved in cancer, such as signal transduction, cell dissociation and invasion, cell–matrix adhesion, angiogenesis, immune regulation and metastasis [20]. As with epithelial cadherin, which plays an essential role in cell–cell adhesion, glycosylation of N-linked glycans on epithelial cadherin in the Golgi apparatus can influence epithelial–mesenchymal transition—which leads to metastatic lesions—through wound healing processes; it is thought that this mechanism may facilitate cancer cells leaving their original site during normal physiological processes like wound healing for metastasis and spread. The GOLPH3 complex plays an essential role in driving tumor progression through various mechanisms, one being its regulation of Golgi glycosylation—essential in promoting cancerous traits and progression [17]. Additionally, the GOLPH3 complex promotes cellular DNA damage response, increasing survival under DNA damage conditions. Furthermore, the complex interacts with components of the retromer complex to increase growth factor-induced mTOR signaling [21]. GOLGA2 or GM130 regulates cell migration by realigning the Golgi apparatus towards its leading edge, while simultaneously playing an essential role in cancer cells through the regulation of Golgi glycosylation and protein transport across membranes. GM130, also plays an essential role by contributing to Golgi glycosylation and membrane trafficking; its downregulation triggers autophagy, reduces angiogenesis and suppresses tumorigenesis—while anomalous Golgi glycosylation has been implicated as both cause and effect in cancer development [22].

This review highlights two Golgi target proteins, GOLPH3 and GOLGA proteins, from this apparatus as they relate to gastroenterological cancers, and they were highlighted in the most works from the literature related to gastroenterological cancers.

## 5. Pathophysiology of GOLPH3 

GOLPH3, also known as GPP34, GMx33 or MIDAS, is a glycoprotein that regulates the traffic between the trans-Golgi network and the membrane of the cell. This glycoprotein has remained expressed in the human genome during evolution, its origins being traced back to yeast [2,23]. GOLPH3 is created by a gene residing on the 5p13 chromosome, a gene that is amplified in the context of different types of carcinomas, for example, hepatocellular cancer and gastric cancer. The GOLPH3 complex consists of GOLPH3, MYO18A and Arf1 proteins which work in concert to regulate Golgi membrane trafficking and function. When this complex becomes disordered, its improper function has been linked to various diseases—most notably cancer, since its malfunction can promote tumor growth through altering Golgi glycosylation pathways as well as other signaling pathways. Targeting this complex has, thus, become an attractive strategy in developing therapies against cancer [24,25]. The GOLPH3 complex, made up of GOLPH3, MYO18A and Arf1, plays an essential role in numerous cellular processes, such as regulation of Golgi glycosylation (Figure 1), DNA damage response enhancement, growth factor-induced mTOR signaling enhancement and cell migration regulation. The GOLPH3 complex and Golgi protein GM130 play an essential role in GOLPH3 glycosylation and membrane trafficking of cancer cells, with reduced expression leading to autophagy, decreased angiogenesis and tumorigenesis suppression when reduced. As such, these are both potential targets for future cancer therapies [25].

According to a study conducted in 2013, GOLPH3 is highly expressed in gastric cancer, with this elevated expression correlated with adverse clinical outcomes such as decreased overall survival and tumor invasion. GOLPH3’s role in gastric cancer progression is also explored, specifically its regulation of Golgi membrane trafficking and glycosylation. The study suggests that targeting GOLPH3 may provide a promising avenue for creating therapeutic approaches to treat gastric cancer. Studies conducted on gastric cancer patients demonstrated that higher GOLPH3 expression levels were related to advanced tumor stages, lymph node metastasis and lower overall survival rates. GOLPH3 expression was also positively correlated with cancer stem cell marker CD44 expression as well as EMT markers associated with increased tumor invasion and metastasis. Researchers conducted in vitro experiments to explore the functional role of GOLPH3 in gastric cancer cells. One team discovered that decreasing GOLPH3 expression led to decreased cell proliferation, migration and invasion while simultaneously increasing apoptosis; additionally, knocking down GOLPH3 resulted in lower expression of EMT markers as well as reduced activation of the PI3K/Akt/mTOR signaling pathway which is frequently dysregulated during cancer progression [24].

Overall, this study highlights the role GOLPH3 plays in gastric cancer progression and suggests it may serve as a potential therapeutic target [24].

## 6. GOLGA Proteins

Gastric cancer involves numerous Golgi proteins such as GM130 (GOLGA2), GOLPH3, golgin-160 (GOLGA4), golgin-97 (GOLGA1) and golgin-84 (GOLGA7) that all play unique roles in cell processes including protein glycosylation, vesicle trafficking and Golgi structure maintenance. Their dysregulation has been associated with gastric cancer onset and advancement. Of these proteins, GM130 and GOLPH3 are associated with cancer cell invasion while golgin-160 and golgin-97 are involved with growth factor receptor transport as well as protein glycosylation and protein glycosylation processes, respectively [26].

GM130 (GOLGA2) is an integral component of the Golgi matrix found within the cis-Golgi compartment that plays an essential role in maintaining its structure and stacking of Golgi cisternae as well as transport and glycosylation of proteins and lipids along the secretory pathway. Knockdown of GOLGA2/GM130 disrupts the uniform distribution of Golgi enzymes, leading to glycosylation defects in secretory and membrane proteins as well as inhibition of lateral cisternal fusion of Golgi stacks. GOLGA2/GM130 deficiency causes partial inhibition or delay of transport from the endoplasmic reticulum to the Golgi apparatus. A mutant CHO cell line lacking GOLGA2/GM130 also exhibits temperature sensitivity of cell growth and glycosylation defects [27,28,29].

Golgin-160, a Golgi-associated protein, has been identified as an integral factor in increasing cell surface expression of beta-1 adrenergic receptor (β1AR). This protein plays an essential role in trafficking and sorting proteins within the Golgi apparatus, with multiple experimental techniques such as co-immunoprecipitation and confocal microscopy showing its interaction with β1AR to facilitate its transport from the Golgi apparatus to the cell surface, increasing surface expression [30].

Depletion of golgin-160 resulted in decreased cell surface expression of β1AR, underscoring its critical role in controlling the trafficking and cell surface abundance of this receptor. This finding had significant ramifications for cell responses to adrenergic signaling as having more available receptors on cell surfaces would impact their interactions with ligands as well as initiate downstream signaling pathways (Figure 2).

Further investigations in this field can lead to an improved understanding of the mechanisms governing adrenergic receptor signaling regulation and open new opportunities for novel therapeutic approaches targeting golgin-160 in diseases with dysregulated β1AR function [31,32].

According to Maag RS. et al. (2005), golgin-160 proteins resistant to caspase proteases prevent the activation of apoptotic signaling pathways, leading to disruptions in cell death processes and suggesting that it may offer some protection from programmed cell death under certain circumstances. These findings deepen our knowledge of the mechanisms regulating apoptosis and highlight golgin-160 as a potential influencer in cell survival upon secretory pathway stress and activation of death receptors. Further investigation can lead to novel therapeutic approaches targeting golgin-160 that modulate cell death processes for diseases wherein cell death regulation is impaired [32].

GOLGA1, commonly referred to as golgin-97, primarily resides in the trans-Golgi network (TGN) where it tethers endosome-derived vesicles for transport through retrograde transport back to the TGN [33,34,35]. Depletion of golgin-97 causes fragmentation of the Golgi apparatus [36]. Additionally, it plays an essential role in poxvirus replication [37].

The trans-Golgi network (TGN) serves as an important sorting hub within the Golgi apparatus, helping direct newly synthesized proteins and lipids towards their intended destinations. Furthermore, it serves as an entry point for retrograde transport of endocytic cargos through which golgin-97 is recruited into this network through interactions with Arl1 [35].

Depletion of golgin-97 disrupts trafficking of Shiga toxin subunit B from early endosomes to the TGN and hampers E-cadherin cargo leaving the TGN. Low levels of golgin-97 expression have also been associated with poor patient survival and increased breast cancer invasiveness [36].

Mechanistic investigations have demonstrated that depletion of golgin-97 leads to an unprecedented drop in IkBa protein levels and activation of NF-kB; this activation can promote cell migration and invasion processes and provide insight into golgin-97’s role in these processes [38].

Golgin-84 is a peripheral membrane protein that resides within the Golgi apparatus, particularly within its cis-Golgi network (CGN) and its cisternae. This protein has various distinct domains including its C-terminal coiled-coil domain, an N-terminal GRIP domain and multiple coiled-coil regions in its central region. Golgin-84 interacts with Rab1 (a small GTPase protein) to support Golgi structure and function. Rab1 regulates membrane traffic between the endoplasmic reticulum (ER) and the Golgi apparatus, with golgin-84 binding to its GTP-bound form to form a complex that facilitates tethering and fusion of vesicles in the Golgi apparatus. Depletion or knockdown of golgin-84 leads to fragmentation and dispersion of the Golgi apparatus, underlining its significance in maintaining its integrity and structure. Loss of golgin-84 disrupts Golgi stack organization, disrupting the compartmentalization and processing of cargo proteins within Golgi stacks [39].

It has also been implicated in the regulation of protein transport between the Golgi apparatus and the ER. Interacting with components of the COPI coat protein complex, which mediates retrograde transport from Golgi to ER, it indicates that golgin-84 may play an integral part in the retrograde trafficking of proteins from the Golgi apparatus back towards the ER [39].

Understanding the role of golgin-84 in Golgi structure and function offers insight into its molecular mechanisms underlying Golgi organization, providing us with new understanding of its molecular processes. In future research, we could investigate its functional significance in cell processes like protein glycosylation, cargo sorting, secretion or its potential role in diseases associated with Golgi dysfunction such as various neurodegenerative disorders or cancer [25,39].

## 7. The Impact of Golgi Proteins in Gastric Cancer

First, GOLPH3 expression levels were evaluated between gastric cancer tissues and adjacent noncancerous ones using immunohistochemical staining and quantitative analysis. Through this technique, it was observed that GOLPH3 expression was significantly higher in cancerous than noncancerous tissues—suggesting its possible connection to the progression of gastric cancer [40].

Golgi proteins implicated in gastric cancer include golgin-160, which promotes cell surface expression of beta-1 adrenergic receptor and can potentially alter responses to adrenergic signaling; golgin-97 plays an integral part in protein trafficking and sorting within the Golgi apparatus and has been associated with poor survival and increased invasiveness in breast cancer. Golgin-84 acts as a binding partner of Rab1 to maintain Golgi structure maintenance. GOLPH3 shows higher expression levels in advanced stages of gastric cancer and has been linked with lymph node metastasis and worse survival rates. It regulates cell proliferation, migration and invasion within gastric cancer cells while activating the PI3K/Akt/mTOR signaling pathway which often becomes dysregulated during cancer progression [7,39,40].

To gain more insight into the functional role of GOLPH3 in gastric cancer, in vitro experiments using gastric cancer cell lines were conducted. Reducing GOLPH3 expression via RNA interference techniques resulted in significant reductions in cell proliferation, migration and invasion—evidence suggesting GOLPH3 is integral for encouraging aggressive behavior of gastric cancer cells [40].

According to a study by Peng J et al. (2014), which explored the molecular mechanisms underlying GOLPH3’s effects in gastric cancer, uncovering positive regulation of PI3K/Akt/mTOR signaling pathway by GOLPH3, this pathway has long been known for affecting cell survival, proliferation and migration processes as well as epithelial–mesenchymal transition (EMT) markers such as E-cadherin and N-cadherin expression associated with increased cancer cell invasiveness [40].

These initial findings provide valuable insight into how GOLPH3 contributes to gastric cancer progression. These data indicate that it promotes tumor aggressiveness by activating key signaling pathways and inducing EMT, making targeting GOLPH3 and its related pathways an appealing therapeutic approach for treating gastric cancer [40].

However, it should be borne in mind that further research and validation are required in order to validate these findings and fully understand GOLPH3’s involvement in gastric cancer progression.

## 8. Golgi Proteins in Liver Cancer

Hepatocellular carcinoma (HCC) is the leading form of primary liver cancer among adults, being responsible for most deaths among cirrhotic patients. HCC can result from various causes, including chronic liver inflammation caused by Hepatitis B and C viruses (HBV and HCV, respectively), the latter of which being the leading cause in North America, Europe and Japan. HCV is a small RNA virus that completes its lifecycle within human cells, predominantly liver cells but also in lymphocytes to a lesser degree [41]. After their expression in the rough endoplasmic reticulum (ER) and proteolytic processing of their precursor polypeptide, viral proteins undergo post-transcriptional modifications such as glycosylation. HCV completes its life cycle by producing its lipid envelope and exiting from host cells via lipoviroparticle maturation in both the Golgi apparatus and trans-endosomal secretory routes. However, an extracellular step must also be completed for HCV maturation to occur successfully. PI(4)P plays an essential role in HCV maturation via regulation of its cycle, while GOLPH3 acts as an effector in the Golgi apparatus for this process [42]. Additionally, HCV particle release depends on ER–Golgi secretory trafficking and TGN function. Ten years ago, researchers discovered the first direct connection between HCV and GOLPH3, showing that its interaction with unconventional myosin MYO18A is essential for vesicle budding, and GOLPH3 deficiency impairs HCV secretion without altering intracellular replication of the virus. Hypotheses suggest that chronic inflammation related to HCV infection, which can eventually lead to HCC, could use GOLPH3 as a mechanism for tumor formation [42,43]. According to this theory, GOLPH3 promotes viral invasion and could provide a target for anti-HCV drugs to suppress HCV’s spread within liver tissue. Notably, however, GOLPH3 and HCC do not share an exclusive connection [43].

GP73 is a transmembrane protein located primarily within the Golgi apparatus of cells. Recently, its expression has become an indicator for liver diseases due to its differential expression across various pathologies [44].

This study explores the diagnostic significance of GP73 in various liver diseases, specifically hepatocellular carcinoma (HCC), viral hepatitis, alcoholic liver disease and non-alcoholic fatty liver disease (NAFLD). For HCC cases specifically, levels of GP73 were found to be significantly elevated compared to healthy individuals or those with non-malignant liver conditions, suggesting it can serve as a non-invasive biomarker for detection and monitoring of HCC, potentially aiding in early diagnosis and prognosis evaluation [45].

GP73 showed remarkable diagnostic potential in distinguishing viral hepatitis infections. It displayed higher sensitivity and specificity than traditional liver function markers such as Alanine Aminotransferase (ALT) and Aspartate Aminotransferase (AST), demonstrating its usefulness as a valuable tool in accurately diagnosing and distinguishing different viral subtypes of hepatitis [46].

Granovsky M. et al. (2000) investigated the role of GP73 in alcohol-induced liver injury. Researchers observed that levels of GP73 increased significantly among those who experienced alcohol-related liver damage when compared with healthy controls, suggesting it can help identify and assess severity [14].

Furthermore, GP73 showed promise in treating non-alcoholic fatty liver disease (NAFLD). Researchers discovered that levels of GP73 were elevated among patients suffering from non-alcoholic steatohepatitis (NASH), an advanced and potentially progressive form of NAFLD, when compared to simple steatosis patients—suggesting it can assist in differentiating between them as part of disease diagnosis and management [14].

Overall, the article highlights GP73’s potential diagnostic value in liver diseases like HCC, viral hepatitis, alcoholic liver disease and NAFLD. Due to its noninvasive measurement methods and superior diagnostic accuracy compared with traditional markers, its inclusion as part of routine clinical practice is extremely attractive; however, further research must be conducted into validating its performance, standardizing measurement methods and exploring its use across other liver conditions [14,47].

A relevant and non-invasive way method to evaluate the fatty liver is through the controlled attenuation parameter (CAP), ultimately resulting in enhanced patient management and improved outcomes for individuals with this type of disease [48].

Notably, successful treatment of HCV infection may have an indirect influence on managing liver cancer. By reaching SVR and decreasing inflammation and fibrosis in liver tissue, risk may be reduced for HCC formation. HCV elimination may also help improve liver function and enhance other treatments for liver cancer such as surgery, chemotherapy or radiotherapy. The interferon-free regimen is widely utilized for the treatment of Hepatitis C Virus (HCV) infection; however, evidence linking these drugs directly with liver cancer treatment remains scant. Its primary goal is achieving a sustained virologic response (SVR), eliminating HCV from the system completely and decreasing risks such as progression to liver cirrhosis or hepatocellular carcinoma [49].

## 9. The Impact of Golgi Proteins in Colon Cancer

Golgi proteins have emerged as key players in the development and progression of colon cancer, their dysregulation affecting various cellular processes.

GM130 (GOLGA2), one of many Golgi proteins gaining widespread interest, has attracted much scrutiny. When looking at colon cancer cells specifically, its expression levels tend to be increased relative to those of normal colon tissues compared with tumor tissues, indicating its possible link to tumor progression. Furthermore, evidence has pointed toward this protein’s involvement in cell proliferation, migration, invasion, structural integrity of Golgi apparatus maintenance as well as transport of proteins within secretory pathways.

According to Baschieri F. et al. (2014), the formation of a complex between GM130 and RasGRF regulates Cdc42 signaling, an essential process in cell polarity and tumorigenesis. Golgi protein GM130 interacts with RasGRF (guanine nucleotide exchange factor for Cdc42) to form this complex and modulate Cdc42 signaling. Researchers investigated the implications of the GM130–RasGRF complex on cell polarity and tumorigenesis. Disruption of this complex led to impaired epithelial cell polarization and tight junction formation as well as increased tumor growth and invasiveness. Finally, in a mouse model of colorectal cancer, disruption resulted in greater tumor growth and aggressiveness [50].

This study illuminates the molecular mechanisms governing regulation of Cdc42 signaling by the GM130–RasGRF complex, specifically within the Golgi apparatus, where this complex localizes to, controlling precise activation of Cdc42 at specific cellular regions and restricting Cdc42 activation to the leading edge of migrating cells, thus supporting cell polarity and directed migration [50].

Other Golgi proteins such as GOLGA1 (golgin-97) and GOLGA7 (golgin-84) have been implicated in colon cancer. GOLGA1 plays a role in trafficking proteins through the Golgi apparatus, with poor patient survival rates and increased invasiveness rates seen in breast cancer patients who use this protein. GOLGA7 helps maintain structure while impacting protein glycosylation processes.

Golgi proteins appear to play an integral role in colon cancer progression by modulating signaling pathways and impacting various cellular processes. Their dysregulation could play a part in increasing tumor aggressiveness, modulating signaling pathways or otherwise altering processes within cells [51].

Vitamin D plays a crucial role in cell growth, differentiation and apoptosis—three components integral to cancer development and progression. Studies on cancer cells implicating vitamin D or its receptor (VDR) as an influencer have implicated it with processes including apoptosis, invasion, inhibition of inflammatory cytokine inhibition and regulation of microRNA regulation; yet, its relationship to cancer causality remains unknown due to inconsistent results that result from a lack of standardization for 25(OH)D testing methods or timing measurements, although vitamin D has a particular relationship with colonic tumor cell diferentiation and organoid plasticity [52,53].

## 10. Discussions

Overall, this review emphasizes the crucial significance of Golgi apparatus target proteins—specifically GOLPH3 and GOLGA proteins—in gastroenterological cancers. These proteins play key roles in protein sorting, trafficking and maintaining Golgi morphology; dysregulation has been linked with tumor progression, suggesting their potential as biomarkers and therapeutic targets for gastroenterological cancers [17].

GOLPH3 has long been identified as an oncogene, implicated in advanced tumor stages, metastasis and poor patient outcomes. GOLPH3 exerts its effect by modulating key signaling pathways such as the PI3K/Akt/mTOR pathway which plays a vital role in cell survival, proliferation and migration—providing targetable precision medicine strategies against gastroenterological cancers.

GOLGA proteins such as golgin-97 and golgin-84 have also been implicated in cancer development. Golgin-97 plays an integral part in protein trafficking, leading to increased invasiveness in breast cancer cases. Golgin-84 contributes to Golgi structure maintenance and protein glycosylation processes, further underscoring their significance in cancer biology. Their dysregulation underscores this point.

Understanding the functions and molecular mechanisms of Golgi apparatus target proteins involved in gastroenterological cancer is of critical importance for designing effective targeted therapies. By exploring their roles in tumor progression, signal transduction and protein modifications, novel therapeutic approaches can be devised that selectively inhibit these processes and open up novel treatment opportunities with tailored approaches.

## Figures and Tables

**Figure 1 cells-12-01823-f001:**
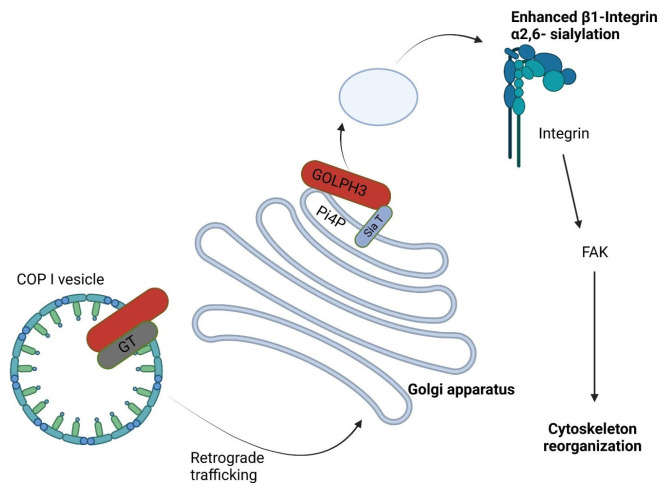
GOLPH3’s role in Golgi protein glycosylation. Several Golgi glycosyltransferases (GT) undergo retrograde trafficking through COPI, which is mediated by GOLPH3. Sialyl-transferase interacts with the GOLPH3 protein and when it is overexpressed, it results in enhanced α2,6-sialylation of β1-integrins. Changes in glycosylation have an impact on the integrin-mediated signaling cascade, which activates FAK (focal adhesion kinase) and causes actin cytoskeleton remodeling and cell movement [4,5,7,23].

**Figure 2 cells-12-01823-f002:**
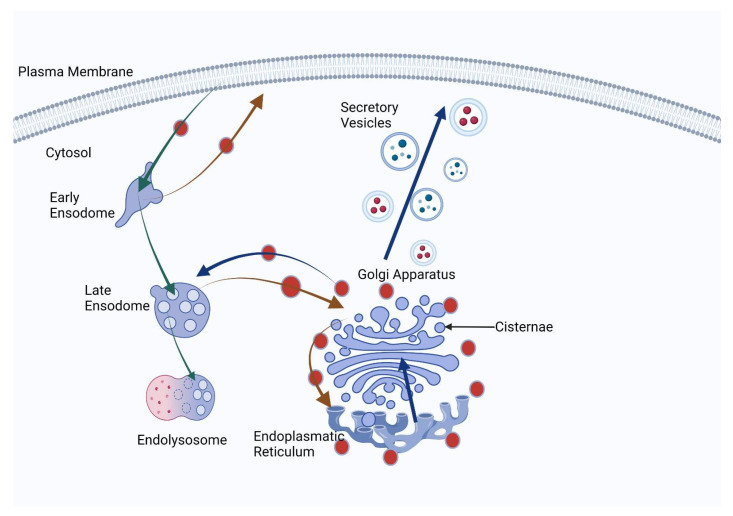
This image illustrates the intricate process of protein processing and transport within the Golgi apparatus. Proteins synthesized in the endoplasmic reticulum (ER) are transported directly to the Golgi apparatus for glycosylation and packaging for export. The Golgi apparatus consists of multiple stacks known as cisternae that contain enzymes responsible for glycan synthesis. These cisternae play an essential role in the assembly and processing of glycan chains which may occur step by step. Once proteins undergo glycosylation, they are carefully packed into secretory vesicles within the Golgi apparatus as transport carriers to reach their final destinations such as plasma membrane, lysosomes or extracellular space. Cells internalize glycans for degradation purposes; the Golgi apparatus ensures their safe transport to lysosomes for degradation. This image provides a vivid depiction of how proteins move through a sequential workflow from transport, modification, packaging and distribution by the Golgi apparatus to glycosylation and final fate of glycosylated proteins or internalized glycans within cells.

## Data Availability

No new data were created.

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
