# Peer review of "Golgi Apparatus Target Proteins in Gastroenterological Cancers: A Comprehensive Review of GOLPH3 and GOLGA Proteins"

_cells, 2023, doi:10.3390/cells12141823_

Round 1

Author Response

Dear reviewer, 

Thank you for your helpful and constructive feedback. Your expertise and insightful comments have guided us in refining the methodology and enhancing the overall presentation of our findings. We have addressed all the comments excluding comment number 2 because we want to keep the introduction as general as possible.

Kind regards, 

The collective of authors

Reviewer 2 Report

This review is nicely written and attractive to the field.

However, I have a few comments

Among all the putative Golgi target proteins in cancers why authors specifically focus on GOLPH3 and GOLGA proteins.

I believe this should be better highlighted.

The figure legend of Figure 1 is not complete. What do the authors want to highlight here is unclear.

Authors should provide an additional Figure with specific contributions within the Golgi of GOLPH3 and GOLGA proteins and how this would be linked to cancer biology.

With the same idea, authors should discuss the role of lipids, calcium, and cytoskeleton in the frame of Golgi proteins in cancer biology since Golgi protein's biological activities rely on the lipids composition of the organelle, calcium, or manganese and cytoskeleton.

Author Response

Dear reviewer,

I am writing to express my sincere gratitude for your invaluable review of our paper. Your thoughtful and constructive feedback has been immensely helpful in improving the quality and clarity of our research. Your expertise and insightful comments have guided us in refining the methodology and enhancing the overall presentation of our findings.

We have addressed all the comments. We modified the images as you suggested.

Kind regards,

The collective of authors

Round 2

Reviewer 1 Report

The revisions are adequate.